# Fungal Infections in Liver Transplant Recipients

**DOI:** 10.3390/jof7070524

**Published:** 2021-06-29

**Authors:** Michael Scolarici, Margaret Jorgenson, Christopher Saddler, Jeannina Smith

**Affiliations:** 1Department of Medicine, University of Wisconsin, Madison, WI 53792, USA; MScolarici@uwhealth.org (M.S.); csaddler@uwhealth.org (C.S.); 2Department of Pharmacy, University of Wisconsin, Madison, WI 53792, USA; MJorgenson@uwhealth.org

**Keywords:** liver transplant, invasive fungal infection, candidiasis, cryptococcosis, *Aspergillus*

## Abstract

Invasive fungal infections (IFIs) are one of the most feared complications associated with liver transplantation, with high rates of morbidity and mortality. We discuss the most common invasive fungal infections in the setting of liver transplant, including *Candida*, *Aspergillus*, and *Cryptococcal* infections, and some less frequent but devastating mold infections. Further, we evaluate the use of prophylaxis to prevent invasive fungal infection in this population as a promising mechanism to reduce risks to patients after liver transplant.

## 1. Introduction

Patients with liver transplants are at enhanced risk of invasive fungal infections (IFIs), which are feared complications with high rates of morbidity and mortality. These infections are common with the incidence of IFIs between 4% and 40% in all liver transplant recipients [1,2]. Underlying liver disease is associated with increased risk of infection from several fungal pathogens, including *Candida* and *Cryptococcal* species, which, in addition to exogenous immunosuppression after transplant, result in unique epidemiology of fungal infections in this population [3]. Patients with liver transplant experience significant morbidity and mortality from IFIs [4,5]. The mortality of IFIs in this population is very high, ranging from 25% to 67% [5]. Here we review the literature regarding IFIs in liver transplant recipients with a focus on the most common and most devastating pathogens.

## 2. *Candida* Species

### 2.1. Epidemiology

*Candida* species are common organisms of pathogenic potential that can colonize the gastrointestinal tract, genitourinary tract, and skin. The major pathogenic species are *C. albicans*, *C. glabrata*, *C. parapsilosis*, *C. tropicalis*, *and C. krusei*. Additionally, *C. auris* is a concerning emerging pathogen that has significant baseline resistance and potential for extensive resistance. The overall invasive fungal infection (IFI) rate post-liver transplant at 1 year is 1.8%, reaching 2.9% at 5 years and 5% by 10 years [6]. Of these IFIs, *Candida* infections represent the vast majority, causing 68–93% of post-liver-transplant IFIs [7,8]. Over time, there has been a trend toward infection with non-*albicans* species. For instance, a Spanish cohort examining candidemia over time in solid organ transplant recipients (SOTRs), comparing 2010–2011 with 2016–2018, showed an increase in *C. glabrata* over time from 18.8% to 30.4% and a decrease in *C. albicans* cases over that period [9]. Further, there has been suggestion of higher mortality with non-*albicans* species. Prior azole use was linked to *glabrata* isolation [10]. Finally, the emerging highly drug-resistant *C. auris* has been reported in liver transplant recipients, including by Theodoropoulos et al., which could threaten both prophylaxis and treatment of invasive candidiasis (IC) in patients with liver transplants [11].

Risk factors for IC in the general population include older age, broad-spectrum antibiotics, central venous catheterization, parenteral nutrition, prolonged neutropenia, prolonged ICU stay, diabetes, renal replacement, and *Candida* colonization [12]. Risk factors specific to liver transplant recipients include anastomotic leak, repeat laparotomy, and choledochojejunostomy [13,14,15]. Factors identified in SOTRs overall include acute renal failure, recent CMV, graft failure, early re-exploration, and *Candida* colonization [16].

### 2.2. Clinical Manifestations

Clinical manifestations are well described elsewhere, but briefly range from asymptomatic colonization to life-threatening infection. The most common forms of invasive disease in liver transplant recipients are candidemia and intra-abdominal infection. It is important to recognize that early post-transplant signs and symptoms of deep *Candida* infection can be nonspecific, and a high index of suspicion is needed.

### 2.3. Diagnosis

The critical step in IC diagnosis is consideration. The gold standard remains culture from a sterile site or identification by histopathology on a tissue sample.

Traditional blood cultures still have significant limitations. In a study involving non-liver-transplant recipients, the sensitivity of blood cultures for the detection of IC was only ~50% [17], and cultures are often negative despite deep-seated infections, such as intra-abdominal infections common after liver transplantation. Other means of detection are necessary.

Beta-D-glucan (BDG) is a component of the fungal cell wall of *Candida* species, as well as other fungi, that can be detected in serum and may be elevated in the setting of *Candida* infection. However, its use is limited by numerous other causes of positivity, such as infection with *Aspergillus* and *Pneumocystis* species, exposure to cellulose membranes for dialysis, platelet infusion with leukocyte-removing filters, immunoglobins, albumin, certain IV antibiotics, bacteremia, surgical gauze containing glucan, and severe mucositis [18]. Sensitivity ranges from 70% to 93%, and specificity from 65% to 87% overall; in a study of liver transplant recipients, two sequential positive BDGs plus evidence of *Candida* colonization had a sensitivity of 83%, a specificity of 89%, and a negative predictive value of 97.6% for IC [19]. This particularly demonstrates the strong negative predictive value of BDG when considering *Candida*, but understanding its test characteristics is critical for appropriate use.

The T2 *Candida* assay has also emerged as a non-culture-based diagnostic test performed directly on whole blood to detect the five most common pathogenic *Candida* species (*C. albicans*, *C. glabrata*, *C. parapsilosis*, *C. tropicalis*, and *C. krusei*) with clinical trial performance of 89–91% sensitivity and 99% specificity. The most attractive feature of this test is that results are available in 3–5 h vs. days with traditional culture. These characteristics have been utilized to improve antifungal stewardship and can reduce time to effective therapy. Currently, it does not detect the emerging highly drug-resistant *C. auris*, but this is in development with promising results [20]. This test is not yet widely available.

PCR-based assays for IC show promise but are not yet readily available for clinical use [21].

Overall, a high index of suspicion is required for the diagnosis of IC, and often a combination of testing approaches is required for adequate evaluation.

Susceptibility testing on *Candida* species is increasingly important, as antifungal pressure is common, and infections with non-*albicans* species are becoming more frequent. Susceptibility testing should be performed in all clinically significant isolates, such as bloodstream or deep cultures, particularly in the setting of prior azole exposure, *C. glabrata*, *C. parapsilosis*, and *C. auris*, as recommended by the Infectious Disease Community of Practice in the American Society of Transplantation (ID AST COP) [22].

### 2.4. Treatment

The mainstays of therapy are early consideration of *Candida* infection, to allow for prompt initiation of therapy, and rapid attainment of adequate source control. Given the frequency of non-*albicans* species, perioperative azole exposure, and nosocomial contact associated with liver transplantation, empiric therapy should typically be an echinocandin. Stepping down to alternative therapy, such as an azole, should be guided by species identification and susceptibility testing. Therapy should follow the Infectious Disease Society of America (IDSA) 2016 guidelines endorsed by ID AST COP [22,23].

### 2.5. Outcomes

Despite ongoing efforts toward prevention and improved management of fungal infection, IC still has significant effects on graft function, morbidity, and mortality post-transplant with reports of 90-day mortality of 26% [7].

Overall, *Candida* infections are the most common post-liver-transplant IFIs, have significant risk of harm, and require a high index of suspicion, rapid diagnostic evaluation which is often multimodal, and aggressive management, including maximization of source control and early appropriate antifungal therapy.

## 3. *Aspergillus* Species

### 3.1. Epidemiology

Patients with liver transplants are at risk of invasive infections with *Aspergillus* species. The incidence of invasive *Aspergillus* infection ranges from 1% to 9% in patients after liver transplantation [2,24,25]. This rate is impacted by several well-defined risk factors, including prolonged surgical time, massive intraoperative transfusion requirement, re-transplantation, steroid-resistant rejection, acute kidney injury, cytomegalovirus (CMV) infection, diabetes, and use of broad-spectrum antibiotics [26,27].

### 3.2. Pathogenesis and Clinical Manifestations

*Aspergillus* species are ubiquitous saprophytic fungi found in the environment, particularly in areas of decaying vegetation. Over 200 *Aspergillus* species have been identified. In the environment, the fungi produce conidia, which can disperse widely in the air. When they are inhaled, they can cause pulmonary infection ranging from a noninvasive fungal ball to invasion and dissemination depending on a variety of host factors. A variety of *Aspergillus* species have been associated with infection in liver transplant recipients. The most common infecting *Aspergillus* species are *Aspergillus fumigatus* (73%), *Aspergillus flavus* (14%), and *Aspergillus terreus* (8%) [24], but rare species are also reported to cause invasive infections with significant morbidity [28].

The majority of invasive *Aspergillus* infections in liver transplant recipients are pulmonary, as the most common exposure mechanism is inhalation of the mold. Dissemination with widespread disease, including involvement of the heart, eyes, muscles, thyroid, and central nervous system (CNS), is well documented [29,30]. Isolated invasive fungal sinusitis from aspergillosis is also reported [31].

### 3.3. Diagnosis

Timely diagnosis of invasive *Aspergillus* infection in liver transplant recipients requires a high index of suspicion. Patients who develop fever, respiratory symptoms, or symptoms of disseminated infection should be evaluated with advanced chest imaging, such as computed tomography (CT) [32]. Plain radiographs may give falsely-negative results and should not be used to exclude pulmonary involvement. Focal symptoms elsewhere, such as the sinuses, should also have appropriate imaging modalities, such as dedicated sinus CT. Characteristic findings of invasive pulmonary aspergillosis (IPA) on chest CT include ground-glass opacities, cavities, and nodules with or without halo [33]. In a study of patients with proven or probable IPA after liver transplant, three main radiological findings were identified: nodules with or without a halo sign, masses, and consolidations in a patchy pattern. A tree-in-bud pattern was seen in 12% (3/25) of patients. The halo sign was seen in 80% of liver transplant recipients within 1 week after the onset of symptoms. The hypodense sign was observed in large nodules or masses, which then became cavitary within a month after the onset of symptoms in almost 70% [33]. Sinus imaging may reveal invasion of local structures and/or mycetoma [34]. It is not possible to distinguish the species causing invasive fungal sinusitis, and rhinocerebral mucormycosis would also be in the differential of these findings. Diagnostic material should be obtained, directed by imaging findings.

Serological markers can be used to augment diagnostic capabilities and should be paired with imaging and culture for the diagnosis of invasive infection. Serum and bronchoalveolar lavage (BAL) galactomannans (GM) are recommended as accurate markers for the diagnosis of invasive *Aspergillus* infections [32]. A GM index of 0.5 or higher is considered positive [35]. BAL GM has been shown to be even more sensitive and provide an earlier positive result than serum testing or traditional culture methods on BAL fluid in solid organ transplant recipients [36]. BDG is less specific in the liver transplant population, likely because of the presence of factors associated with false-positive testing as listed above. In one study of liver transplant patients on antifungal prophylaxis, a baseline of 50% of the patients tested positive for BDG. In this study, the sensitivity, specificity, positive predictive value (PPV), and negative predictive value (NPV) of BDG for IFI were 75% (95% CI: 65–83), 65% (62–68), 17% (13–21), and 96% (94–97), respectively. The authors proposed that raising the cutoff for positivity to 80 pg/mL enhanced the specificity without sacrificing the sensitivity [37]. In summary, the BDG has good negative predictive values, but the high rate of false-positive tests suggests that it should be interpreted in the complete clinical context [32,35,37].

### 3.4. Treatment

Modern treatment regimens for invasive *Aspergillus* infections are associated with reduced mortality overall, and in particular, the introduction of triazole therapy has dramatically improved outcomes [24]. One of the challenges of treatment of invasive fungal infection in liver transplant recipients is that nearly all available antifungal drugs have some risk of hepatotoxicity. This risk is compounded by the concomitant use of interacting medications, specifically immunosuppressive therapy. Before 2000, the vast majority of patients with IPA after liver transplant were treated with amphotericin B. After the introduction of voriconazole, this agent quickly became the drug of choice. Combination therapy, such as adding an echinocandin to an azole, is rarely used but may be associated with improved outcomes in very select cases [24].

### 3.5. Outcomes

Invasive *Aspergillus* infection remains morbid in liver transplant recipients. In a 2015 study, overall mortality was 66%, with a 1-year overall probability of survival of only 35%. The use of modern triazole therapy, specifically voriconazole, was associated with improved outcomes [24]. Risks of poor outcome of *Aspergillus* infection included short time from transplant to diagnosis (less than 25 days), renal failure and need for dialysis especially, and multiorgan involvement [24].

## 4. *Cryptococcal* Species

### 4.1. Epidemiology

*Cryptococcus* species are ubiquitous fungi found worldwide in soil and bird droppings [38]. Several species are clinically significant, including *Cryptococcus neoformans* and *Cryptococcus gattii*, which are now divided into separate species. Both species cause invasive disease in humans, although their manifestations have some important differences. *C. neoformans* can cause infection in both immunocompromised and apparently immunocompetent patients but tends to behave more like a true opportunist [38]. *C. gattii*, on the other hand, has been associated with cryptococcosis in immunocompetent persons in subtropical regions and more recently the northwest coast of North America, specifically near Vancouver Island and China [39]. When SOTRs become infected with *C. gattii*, they are more likely to have disseminated infection, and the incidence of CNS disease and mortality is much higher [40]. *Cryptococcal* infection is the third most common invasive fungal infection in recipients of solid organ transplant [41] with an overall incidence of 0.3–5% [7,42] and is associated with significant morbidity and mortality. A majority of patients (54–62%) with cryptococcosis after solid organ transplant develop CNS disease [43].

### 4.2. Pathogenesis and Clinical Manifestations

*Cryptococcal* disease is typically associated with inhalation of poorly encapsulated yeast cells or basidiospores from the environment. These establish infection in the lungs, which is often asymptomatic, even in immunocompromised patients. From there, cryptococcus can travel throughout the body, including the CNS, by direct invasion but also in part because they are carried by macrophages in a “Trojan horse”-like mechanism into protected sites [38,44]. Patients with end-stage liver disease (ESLD) are at enhanced risk of developing *Cryptococcal* disease both pre- and post-transplant as a result of physiological changes induced by ESLD, including impaired cell-mediated immunity, phagocyte dysfunction, complement deficiency, and hypogammaglobulinemia [4,45]. Dissemination is much more likely to occur in patients with deficient cell-mediated immunity. Among SOTRs, the risk of developing disseminated disease was significantly higher for liver transplant recipients (adjusted hazard ratio (HR), 6.65; *p* = 0.048) in [46].

*Cryptococcus* can also occasionally be inoculated directly into the skin, causing local infection [47]. The majority of post-transplant *Cryptococcus* infections are from reactivation of latent infection, but acquisition of the fungi after transplant is also reported, including from pet birds [48]. There are also cases of donor origin, but this appears to be rare [49].

Invasive *Cryptococcus* infection may cause lung masses, pneumonia, and fungemia and has a predilection for CNS infection, including mass lesions and meningitis. For that reason, *Cryptococcal* disease should be considered in liver transplant recipients with fever, headache, subacute mental status changes, or mass lesions in the lungs or CNS [41].

### 4.3. Diagnosis

Diagnosis of *Cryptococcal* infection is typically made by *Cryptococcal* antigen (CrAg) assay from serum and cerebrospinal fluid (CSF) or biopsy with fungal stains and cultures [41]. Several assays are available for *Cryptococcal* antigen testing, including latex agglutination (LA), enzyme immunoassay (EIA), and lateral flow *Cryptococcal* antigen assay (LFA). The analytic sensitivity of the LFA was demonstrated to be consistently superior to those of the LA or EIA across all serotypes in [50].

Serum *Cryptococcal* antigen is positive in 88–91% cases of CNS disease [43]; however, in transplant recipients CNS disease may be present even in the absence of detectable antigenemia. CNS antigen testing nearly always yields a positive result in CNS infection. Lumbar puncture should be performed in all transplant recipients with *Cryptococcal* infection to evaluate for CNS disease. Opening pressure should always be obtained as management of elevated pressure is critical for good outcomes. Characteristic CSF findings include mild pleocytosis (20 to 200 cells/µL), mildly elevated protein, and hypoglycorrhachia. The absence of pleocytosis and chemical changes does not rule out CNS involvement, however.

### 4.4. Treatment

New guidelines for the treatment of *Cryptococcal* infections after solid organ transplant has just been released by the ID AST COP [41]. These guidelines are generally in agreement with the IDSA guidelines [37]. Patients with CNS infection, including cryptococcomas and meningitis, and moderate to severe pulmonary infections should be treated with liposomal amphotericin B, plus flucytosine. Management is often complicated by acute kidney injury and cytopenias, so therapeutic drug monitoring (TDM) of the flucytosine is a recommended guideline [37]. If TDM cannot be performed, careful monitoring of renal function and cell counts is important to identify toxicity. After induction, most patients may be treated with prolonged fluconazole maintenance therapy. Patients with mild pulmonary disease do not need induction therapy and may be treated safely with fluconazole throughout the treatment course [41,51].

Infections with *C. gattii* may be associated with higher fluconazole MICs [40]. These infections may require other azoles for step-down therapy based on antifungal susceptibility testing, and optimal therapy in this setting is not well defined.

Reduction of immunosuppression may be helpful in controlling the infection, but rapid reduction in immunosuppression has also been associated with immune reconstitution inflammatory syndrome (IRIS) [52]. This may mimic worsening infection, but since it is treated with anti-inflammatories rather than anti-infectives, it is important to consider whether patients suddenly worsen despite negative cultures. The treatments of IRIS after *Cryptococcal* infection are typically steroids, but newer agents, such as adalimumab, have been used successfully in a refractory case [53]. New guidelines for the treatment of *Cryptococcal* infections after solid organ transplant have just been released [41].

### 4.5. Outcomes

The overall 90-day mortality from *Cryptococcal* infection after solid organ transplant was 14% and was higher among patients with renal failure, abnormal mental status, fungemia, and disseminated disease [46].

Recent data suggest that *Cryptococcal* infection should not be considered an absolute contraindication to proceeding with liver transplantation, but that this decision should be made cautiously as mortality is high in this patient population [4].

## 5. Other Opportunistic Fungi: *Scedosporium, Mucormycetes, Fusarium*

### 5.1. Epidemiology

Invasive molds, such as *Scedosporium*, *Mucomycetes*, and *Fusarium* species, are uncommon after liver transplantation given the lower net immunosuppressive burden when compared with hematopoietic stem cell transplantation (HSCT) or other solid organ allograft subtypes. However, incidence is increasing, likely due to the availability of more potent immunosuppression along with the use of newer antifungal agents for prophylaxis of *Candida* infection after transplantation [54]. Indeed, *Scedosporium* species are increasingly important pathogens in the immunocompromised host, accounting for 25% of non-*Aspergillus* mold infections in transplant recipients [55]. The incidence of mucormycosis in liver transplant recipients has been estimated to be 4–16 per 1000 patients [56,57]. *Fusarium* typically manifests as a local infection after solid organ transplant with disseminated fusariosis being exceedingly rare. In one review of the literature, only six cases were reported between 1979 and 2020 [58].

### 5.2. Pathogenesis, Clinical Manifestations, and Diagnosis: Scedosporium

*Scedosporium* is a soil-dwelling saprophyte. It can also be found in water bodies that have been polluted by environmental contaminants or sewage. There are three pathogenic species that result in human infection, including *Scedosporium apiospermum* and its teleomorphs *Pseudallescheria boydii*, *Lomentospora prolificans* (formally *Scedosporium prolificans*), and *Scedosporium aurantiacum* [59]. When cultured on standard culture media, *Scedosporium* species appear as branching septate hyphae. Infection can occur in varying degrees of severity and typically manifests as pneumonia. Infection can also involve sinuses, bones, joints, eyes, and the CNS. The mode of transmission is typically considered to be spore inhalation; however, disseminated invasive infection after near drowning can occur with *Scedosporium* species being the most common cause of fungal infection in this setting [60]. Additionally, donor-derived infection of the recipient after near drowning of the donor has been described [61].

#### Treatment and Outcomes: *Scedosporium*

Treatment of *Scedosporium* species is challenging. It is intrinsically resistant to amphotericin deoxycholate and all liposomal formulations. *S. apiospermum/P. boydii* may respond to antifungal triazoles, particularly voriconazole, which has the best in vitro profile with MICs of 0.12 to 0.5 mcg/mL in clinical isolates obtained from lung transplant recipients [62]. Due to poor efficacy of single agents, synergy has been explored, with combinations of terbinafine and micafungin with voriconazole, resulting in the best in vitro responses [61,63]. Granulocyte-macrophage colony-stimulating factors have also been used to improve response, given reliance on innate immunity, particularly the role of polymorphonuclear cells. Infections caused by *L. prolificans* do not usually respond to antifungal therapy alone and often require surgery and reversal of immunosuppression [59]. Outcomes in transplant recipients are poor, with mortality estimated to be 59% across species, and mortality attributed to *L. prolificans* and *S. aurantiacum* >70% [61].

### 5.3. Pathogenesis, Clinical Manifestations, and Diagnosis: Mucormycetes

*Mucormycetes* are a class of ubiquitous soil-dwelling saprophytic fungi that are found in decaying plants or animal matter. Most often the isolated genera include *Rhizopus*, *Rhizomucor*, *Mucor*, and *Absidia.* On tissue histopathology, they appear as broad, ribbonlike, sparsely septate hyphae with right angular branching [64]. Although more commonly described in the setting of neutropenia or hyperglycemia related to diabetic disease, high steroid burden, as used to treat rejection, is a postulated risk factor for disease in SOTRs [65,66]. Manifestations range from local cutaneous infection to hematogenous dissemination, with the most common presentation being rhinosinusitis, with or without rhinocerebral extension, followed by pulmonary mucormycosis [65].

#### Treatment and Outcomes: *Mucormycetes*

Treatment requires a multipronged approach with immunosuppressive reduction, surgical invention, systemic amphotericin B, and even local irrigation with amphotericin B [65,67]. In one report, mortality was 50% for any manifestation and was not different across allograft subtypes [65]. Prognoses associated with rhinocerebral and disseminated diseases were poor at 93.3% and 100%, respectively [65]. Survival was improved in patients who had aggressive reduction or discontinuation of immunosuppression compared with those without a change in their regimen (69.5% vs. 46.1%, *p* = 0.05) [65].

### 5.4. Pathogenesis, Clinical Manifestations, and Diagnosis: Fusarium

*Fusarium* species are also common soil-dwelling saprophytes. Human pathogenic species are mainly limited to the *F. solani* complex (*Neocosmospora* species), but also include *F. oxysporum*, *F. verticillioides,* and *F. proliferatum* [68]. Local fusarial infections can occur in both immunocompetent and immunocompromised hosts and are typically the result of trauma. Disseminated disease relies on the presence of neutropenia and impaired macrophage function. Fusarial onychomycosis is typically indolent in the immunocompetent host but can lead to dissemination in the setting of systemic immunosuppression. Identification of fusarium on tissue pathology reveals fine, acutely angular, and dichotomously branching septate hyphae, which are not easily distinguished from other hyaline molds. *Fusarium* species can be isolated on culture media and appear as fluffy/cottony colonies after 2–5 days of growth [58].

#### Treatment and Outcomes: *Fusarium*

*Fusarium* species are resistant to most available antifungal agents. Fluconazole, itraconazole, flucytosine, and echinocandins have no activity. Ketoconazole, miconazole, and terbinafine have only limited activity. Voriconazole and posaconazole have moderate activity with respective MICs of 2–8 mg/L and 0.5–8 mg/L depending on species. Amphotericin B is the most reliably effective agent, with a MIC of 2 mg/L across species, but with relatively poor in vivo efficacy [68,69]. A combined approach of amphotericin B and surgical debridement is typically required for clinical cure [70]. The addition of topical nystatin has had success in burn patients [71]. In a case report involving a combined heart–liver transplant recipient that included a review of the literature, *Fusarium* infections in SOTRs were mainly localized, with fungemia described as “uncommon” compared with 20–60% in HSCT recipients. Additionally, SOTR infection occurred later, mainly >9 months post-transplant compared with early post-transplant in HSCT, and was associated with lower mortality, 33% SOTR vs. 70–100% HSCT [72].

## 6. Endemic Fungi: *Histoplasma*, *Blastomyces*, *Coccidioides*

### 6.1. Epidemiology

*Histoplasma* species and *Blastomyces* species are endemic to the Ohio and Mississippi River Valleys, and *Coccidioides* species are endemic to the southwestern states [73,74,75,76]. Outside these regions, cases may represent remote travel, reactivation, or donor-derived infection particularly within the first month after transplantation [75,77]. Donor-derived blastomycosis has not been reported [78]. Routine screening and prophylaxis are only recommended for coccidioidomycosis in its endemic region [75].

Endemic mycoses in SOTRs are rare with overall incidence estimated to be 0.2% [73]. The incidence of histoplasmosis in a TRANSNET study was 0.102%. In a single center in Wisconsin, the incidence of blastomycosis was 0.27%, 18 times higher than in the general population [73,78]. Therefore, most guidance on endemic mycoses in liver transplant recipients is extrapolated from cohorts evaluating all SOTRs. Time from transplantation to infection is bimodal; in the TRANSNET study, 40% of 64 cases were diagnosed within the first 6 months, and 34% ranging from 2 to 11 years [73]. This pattern was seen in other cohorts [78,79].

### 6.2. Pathogenesis and Clinical Manifestations

Aerosolized conidia and spores, typically inhaled with exposure to disrupted soil, convert from mold to yeast forms at body temperature capable of dissemination [80,81,82]. Presentation ranges from indolent pulmonary infection to acute respiratory distress syndrome (ARDS) and extrapulmonary dissemination to the skin, osteoarticular system, CNS, and genitourinary system [75]. SOTRs are at increased risk of severe disease, dissemination, and death [75,76,78,79]. The rates of dissemination in SOTRs were 81% for histoplasmosis [79], 75% for coccidioidomycosis [77], and 37.5% for blastomycosis in [78]. Blastomycosis disseminates at similar rates regardless of immune status, but SOTRs are more likely to have severe disease, 84.2% vs. 47.3% [78].

### 6.3. Diagnosis

The gold standard for diagnosis is isolation on culture, but distinctive histopathology or direct microscopy of affected sites proves infection. *Histoplasma* antigen or *Blastomyces* antigen in urine, serum, or body fluid; *Coccidioides* antibodies in CSF; and two-fold rise in *Coccidioides* serum antibodies support probable infection [74,83]. In general, the sensitivity of the antigen EIA is higher for urine samples and disseminated disease. For isolated pulmonary histoplasmosis, sensitivity is 65% for urine, 69% for serum; and for disseminated disease, 90% for urine, 80% for serum [74]. For blastomycosis, sensitivity ranges from 76% to 90% and 56% to 82% for urine and serum, respectively [74]. The EIA antigen tests are specific for endemic mycosis, but there is significant cross-reactivity between histoplasmosis and blastomycosis [74]. Although serologic studies are less reliable for immunosuppressed SOTRs, the highly sensitive coccidioidomycosis EIAs for IgM and IgG are used for screening, and results are confirmed with a more specific and quantitative complement fixation test [76].

### 6.4. Treatment

Treatment of severe or non-CNS disseminated histoplasmosis or blastomycosis should start with 5 mg/kg/day of intravenous liposomal amphotericin B for the first 1–2 weeks or until clinical improvement, and with CNS involvement, it should extend to 4–6 weeks [75,84]. This is followed by 12 months of azole therapy, typically itraconazole, 200 mg, three times daily for 3 days, then 200 mg, twice daily for 12 months [75,84,85]. Fluconazole, 400 mg, daily for 12 months, is the first line for coccidioidomycosis, but CNS dissemination may require 800–1200 mg daily, and severe disease requires intravenous amphotericin B [75,86]. Therapeutic drug level monitoring of azole treatment is recommended, particularly given interactions with calcineurin inhibitors [75]. Relapsed histoplasmosis is associated with persistent urinary antigen at the end of therapy, ≥2.0 ng/mL, and failure to reduce calcineurin inhibitor immunosuppression [79]. If immune suppression is not reversible, then suppressive therapy should be considered [84,85,86]. For CNS disseminated coccidioidomycosis, suppressive therapy should be lifelong regardless of immune status [86].

### 6.5. Outcome

Mortality inclusive of all endemic mycoses in the TRANSNET study was 16% [73]. In separate cohorts, mortality was 10% for histoplasmosis [79], 30–50% for coccidioidomycosis [76,87], and 21.2% for blastomycosis [78]. The risk of death from histoplasmosis was associated with increased age, severe disease, fungemia, and higher urine antigen [79]. Blastomycosis mortality was highest, 66.7%; in cases with ARDS after diagnosis was delayed by two or more courses of antibiotics [78].

## 7. Prophylaxis of Fungal Infection

As we have shown, liver transplant recipients experience invasive fungal disease from a wide range of etiologies, but with infections due to *Candida* species being predominant, and *Aspergillosis* being the second most common etiology.

With these patterns, antifungal prophylaxis has been an attractive option, although the ideal approach remains unclear. This is further complicated by evolving eras, with the availability of new antifungals, changes in patient characteristics and management strategies over time, and more prevalent non-*albicans, Candida* infections. Prophylaxis strategies toward *Candida* and *Aspergillus* have been tested, and recommendations exist for both.

Regarding *Candida* prevention, studies have demonstrated the safety and efficacy of universal prophylaxis with reduction in IFI, although without mortality benefit [88,89,90,91]. Lack of mortality benefit, as well as a high number needed for treatment, has driven interest in targeted prophylaxis. This approach has shown efficacy, reducing IFI, and safety in numerous studies with a variety of agents, including fluconazole, amphotericin B, voriconazole, and echinocandins [92,93,94,95,96,97,98,99,100].

The ID AST COP *Candida* guidelines [22] define high risk as any one of the following: retransplantation, reoperation, renal failure requiring dialysis, transfusion ≥40 units of cellular blood products including autotransfusion, choledochojejunostomy, and *Candida* colonization perioperatively. They also list considerations for MELD ≥30, biliary leaks, and living liver transplant.

Just as the indications and agents studied vary, so does duration. Studies range from 5 days to 10 weeks. We recommend continuation through 2–4 weeks or until discharge if sooner. As noted, the ideal agent is unclear, but given its availability, tolerability, and excellent efficacy in clinical trials, we would recommend fluconazole as the first-line agent. Consideration of echinocandins is appropriate if the risk for fluconazole-resistant *Candida* colonization is felt to be very high due to preceding azole exposure or known resistant *Candida* colonization.

Overall, targeted *Candida* prophylaxis in high-risk recipients is safe and efficacious in decreasing post-transplant IC, and protocols to ensure its appropriate utilization can improve outcomes [101].

A more difficult decision concerns *Aspergillus* prophylaxis in liver transplant recipients. Although the second most common IFI post-liver transplant, it remains relatively rare with rates varying from 1% to 9%, although rates as low as <1% have also been reported [102]. The infrequent but potentially devastating nature suggests that targeted prophylaxis may be ideal, but to date, data are suboptimal.

Currently, the IDSA does not make specific recommendations on *Aspergillus* prophylaxis in liver transplant recipients, and recommending institution-specific guidelines based on local epidemiology and noting optimal duration of prophylaxis are unknown [32]. However, the ID AST COP released guidelines in 2019 recommending targeted prophylaxis in cases of retransplant, renal replacement therapy at the time of or within 7 days of transplant, and reoperation involving the thoracic or intra-abdominal cavity. They recommend echinocandin or voriconazole for targeted prophylaxis, with lipid formulation amphotericin B as alternative, with a duration of 14–21 days [102].

Neyra et al. performed a retrospective study evaluating the performance of risk factors for invasive mold in 534 liver transplant recipients from 2010 to 2014. Cases were rare, with 0.78% of those with risk factors infected vs. 0.98% of those without risk factors infected. Of the 18% of cases with risk factors receiving mold prophylaxis, none suffered mold infection. This study highlights the low sensitivity of traditional mold risk factors, and overall infrequent occurrence hints at the potential benefit of mold prophylaxis and emphasizes that better methods for targeting prophylaxis are needed [103].

Robust data on the optimal agent or duration of prophylaxis are lacking. A trial on universal voriconazole prophylaxis vs. targeted prophylaxis demonstrated good tolerance and no difference in IFI rates, suggesting that targeted prophylaxis allows for the safe reduction of antifungal exposure [93]. Evaluation of echinocandins vs. fluconazole prophylaxis has failed to demonstrate clear superiority, although one trial did show lower rates of *Aspergillus* identification post-liver transplant, 3% vs. 9% when utilizing anidulafungin vs. fluconazole [104,105]. A retrospective evaluation of caspofungin vs. no prophylaxis across eras at a single center found a decrease in IPA diagnoses at 90 days post-transplant, 0.5% vs. 3.3%, suggesting possible efficacy [106]. Balogh et al. retrospectively evaluated high-risk patients treated with voriconazole prophylaxis and found no breakthrough invasive *Aspergillus* episodes and relatively good tolerance [107].

Recently, a retrospective evaluation of *Aspergillus*-colonized liver transplant recipients, all of whom received *Aspergillus*-directed antifungal prophylaxis, predominantly with voriconazole, found only one death due to fungal infection, but also noted decreased survival from baseline, and breakthrough infections were identified. The median duration of prophylaxis was also prolonged, 85 days vs. typical 8 weeks [108]. This suggests that transplantation can be sought with *Aspergillus* colonization but carries increased risk, and that any pretransplant colonization demands prophylaxis and expert management.

Overall, targeted *Aspergillus* prophylaxis remains controversial due to lack of high-quality prospective trials, suboptimal performance of risk factors, and likely regional and temporal variation of baseline risk. Mold prophylaxis appears to be at least safe and may be appropriate based on ID AST COP recommendations, but further prospective trials are necessary. Currently, institution-specific approaches based on local trends seem most appropriate.

## 8. Conclusions

Progress in the identification, prophylaxis, and treatment of IFI after liver transplantation has resulted in encouraging trends in the incidence, morbidity, and mortality from these dreaded infections. Continued research to help identify patients most at risk and the most effective prevention is needed.

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
