# Peer review of "Fungal Infections in Liver Transplant Recipients"

_jof, 2021, doi:10.3390/jof7070524_

Round 1
Reviewer 1 Report
This study is about IFI in liver transplant patients. The authors have reviewed a very wide range of topics relatively adequately, but need to reconsider the following areas:
The introduction needs to explain the background of this review. The background is not an abstract repetation. Considering the incidence, it is better to describe Candida, Aspergillus, and Cryptococcus in order.
In line 19, the fact that the incidence of the IFI at Line 19 is 40% is irrelevant to what follows. The suggestion that prophylaxis is required is not appropriate in the introduction .
In line 31, a comma is required between genus and species. In lines 43,44,46, the first notation of abbreviations is necessary to describe them originally.
You need a reference from Line 86.
In line 95, empirical antifungal therapy must be specified.
In line 102, abbreviation requires description in its original language.
In line 125, the pathogenesis of aspergillosis is lacking.
In line 135, CT findings are additionally required, and images need to be added if necessary.
In line 144, it is necessary to describe the sensitivity, specificity, and cut-off of GM, and it is better to present data from LT patients as well.
In line 158, to my knowledge, combination therapy is uncommon among IPA patients. Please avoid exaggerating narratives.
In line 174, considering the incidence, it would be better to put neoformans first and gatti later. Description of reference description of this sentence is needed. The relatively detailed description of C. gatti infection of immunocompromised hosts is known as a study conducted in China.
In 186, was there any reason to deal with the pathogenesis of cryptococcus in detail compared to candida or aspergillosis? In the case of clinical manifestion, it is necessary to describe content specific to LT.
In line 210, as far as I know, it is not a common test in LFA, nor is it a standard diagnostic modality.
In line 215, additional CSF findings as well as opening pressure is needed.
In line 220, TDM of flycosine is not a commonly available method. It is advisable to only describe the side effects of FC drugs.
In 234, it is necessary to introduce the content that the author considers important.
In 273, it will be necessary to add a reference of GM-CSF.
In 282, the following sentence appears to be related to diagnosis, not pathogenesis or clinical manifestation.
In Prophylaxis, it is necessary to specifically introduce the protocol of prophylaxis suggested by some research groups.
In prophylaxis, it would be necessary to describe specific details about the combination of active monitoring such as BDG or serum GM.
Author Response
Reviewer 1;
We sincerely appreciate your careful review. We have addressed your concerns as below.
This study is about IFI in liver transplant patients. The authors have reviewed a very wide range of topics relatively adequately, but need to reconsider the following areas:
The introduction needs to explain the background of this review. The background is not an abstract repetation. Considering the incidence, it is better to describe Candida, Aspergillus, and Cryptococcus in order.
Thank you. We have rewritten this section which will increase clarity for the reader.
In line 19, the fact that the incidence of the IFI at Line 19 is 40% is irrelevant to what follows. The suggestion that prophylaxis is required is not appropriate in the introduction .
Thank you for noting this, we have rewritten this section.
In line 31, a comma is required between genus and species. In lines 43,44,46, the first notation of abbreviations is necessary to describe them originally.
Thank you for this suggestion. The piece has been updated throughout with the correct nomenclature.
You need a reference from Line 86.
Thank you for your careful review. We apologize for this error. The reference was added.
In line 95, empirical antifungal therapy must be specified.
Thank you for your suggestion which will increase clarity for the reader. This section was updated per your recommendation.
In line 102, abbreviation requires description in its original language.
Thank you for your suggestion which will increase clarity for the reader.
In line 125, the pathogenesis of aspergillosis is lacking.
Thank you for your suggestion which will increase clarity for the reader. This section was rewritten.
In line 135, CT findings are additionally required, and images need to be added if necessary.
Thank you for your suggestion which will increase clarity for the reader. This section was rewritten.
In line 144, it is necessary to describe the sensitivity, specificity, and cut-off of GM, and it is better to present data from LT patients as well.
Thank you for noting this, we have rewritten this section.
In line 158, to my knowledge, combination therapy is uncommon among IPA patients. Please avoid exaggerating narratives.
Thank you for your careful review. We apologize for this error. It was corrected.
In line 174, considering the incidence, it would be better to put neoformans first and gatti later. Description of reference description of this sentence is needed. The relatively detailed description of C. gatti infection of immunocompromised hosts is known as a study conducted in China.
Thank you for noting this, we have rewritten this section.
In 186, was there any reason to deal with the pathogenesis of cryptococcus in detail compared to candida or aspergillosis? In the case of clinical manifestion, it is necessary to describe content specific to LT.
Thank you for noting this, we have rewritten this section and added pathogenesis sections.
In line 210, as far as I know, it is not a common test in LFA, nor is it a standard diagnostic modality.
Thank you for noting this, we have rewritten this section.
In line 215, additional CSF findings as well as opening pressure is needed.
Thank you for your suggestion which will increase clarity for the reader. This section was updated per your recommendation.
In line 220, TDM of flycosine is not a commonly available method. It is advisable to only describe the side effects of FC drugs.
Thank you for noting this. While TDM is recommended in the IDSA guidelines we have amended this section.
In 234, it is necessary to introduce the content that the author considers important.
Thank you for your suggestion which will increase clarity for the reader. This section was updated per your recommendation.
In 273, it will be necessary to add a reference of GM-CSF.
Thank you for your careful review. This sentence was updated with the appropriate reference.
In 282, the following sentence appears to be related to diagnosis, not pathogenesis or clinical manifestation.
Thank you for noting this, section headers were updated to include “diagnosis” as pathology was described for each organism.
In Prophylaxis, it is necessary to specifically introduce the protocol of prophylaxis suggested by some research groups.
Thank you for your suggestion which will increase clarity for the reader. This section was updated per your recommendation, specifically these protocols were summarized for clarity and brevity.
In prophylaxis, it would be necessary to describe specific details about the combination of active monitoring such as BDG or serum GM.
Thank you for your suggestion. We feel that the poor specificity of these tests limits their value and it is therefore not recommended to use them so we did not include them in our discussion.
Reviewer 2 Report
The authors give an overview of the most common fungal infections in liver transplant recipients. Overall the manuscript is well written with a clear structure. The objective fits the focus of the readers of this journal.
In general, the manuscript could be improved by focussing on liver transplant recipients, what is known for this patient group and what is different when you compare this group to other SOT of hemato-oncology patients regarding diagnosis, clinical presentation and outcome. It should be very clear when reading the manuscript to which populations the outcome rates apply: are these liver transplant recipients, SOT recipients (and what organs, long/heart, kidney?) or outcome rates in general (some examples in general comments)?
Comments
- The abstract is a just copy of the introduction and should be rewritten
- throughout the manuscript, species should be writen as C. albicans C. glabrata ((note the point after C. and italics. glabrata without C. is wrong (e.a. line 39 and line 42)
- Many abbreviations are not defined. OLTx line 43, ID AST COP line 94, IPA line 138, abx line 46, line 193: define ESLD
- Line 147 here is 1,3)-β-d-Glucan used : previously just beta-D-glucan or BDG is used
-
line 137-139" Characteristic findings of IPA on chest CT include ground glass opacities, cavities or nodules with or without 138 halo [33]. "Are the CT characteristics different then neutropenic patients?
- line 105-107 "Despite ongoing efforts toward prevention and improved management of fungal in-105 fection, IC still has significant effects on graft function, morbidity, and mortality post-transplant with reports of 90 day mortality of 26%." In liver transplant recipients or in general?
-
line 146: "(1,3)-β-d-Glucan is less specific in the liver transplant popula-146 tion than other high-risk populations such as hematologic malignancy, allogeneic HSCT 147 and both tests should be interpreted in the complete clinical context [32, 35, 36]. " Why is it less specific
- line 161 : Here the first time "disseminated" aspergillosis is used.
-
Line 163. "poor outcome was 2.5 time higher if multiorgan involvement was present."Thus ~85% mortality in multi organ involvement? Please clarify and rephrase
- line 177 * C. gattii
-
Paragraph diagnosis, line 209. Questions: cryptococcal antigen on what sample: serum/CSF fluid? Any specific sensitivity/specificity for liver transplant patients. Is the sensitivity of 88-91% for liver transplants only or all SOT? was the lateral flow assay used or the (less sensitive) ELISA?
-
Outcome paragraph cryptococcosis. Nothing is mentioned about outcome. What is known for liver transplant patients.
-
zygomycetes should be mucormycetes.
- line 247 Should be Scedosporium and Lomentospora based on the reference?
-
Line 250-251 "Fusarium typically manifests as local infection after solid organ transplant with fusariosis being exceedingly rare" Probably more clear to state disseminated fusariosis?
-
Line 257. Scedosporium prolificans (Lomentospora prolificans) should be Lomentospora prolificans (previously Scedosporium prolificans)
-
line 368: liposomal or conventional.
Author Response
Reviewer 2
The authors give an overview of the most common fungal infections in liver transplant recipients. Overall the manuscript is well written with a clear structure. The objective fits the focus of the readers of this journal.
In general, the manuscript could be improved by focussing on liver transplant recipients, what is known for this patient group and what is different when you compare this group to other SOT of hemato-oncology patients regarding diagnosis, clinical presentation and outcome. It should be very clear when reading the manuscript to which populations the outcome rates apply: are these liver transplant recipients, SOT recipients (and what organs, long/heart, kidney?) or outcome rates in general (some examples in general comments)?
We sincerely appreciate your careful review. We have addressed your concerns as below to the best of our abilities. In some cases, we are limited by the content of the primary literature, in that specific data regarding liver transplant recipients may not be available.
Comments
- The abstract is a just copy of the introduction and should be rewritten
Thank you. We have rewritten this section which will increase clarity for the reader.
- throughout the manuscript, species should be writen as C. albicans C. glabrata ((note the point after C. and italics. glabrata without C. is wrong (e.a. line 39 and line 42)
Thank you for this suggestion. The piece has been updated throughout with the correct nomenclature.
- Many abbreviations are not defined. OLTx line 43, ID AST COP line 94, IPA line 138, abx line 46, line 193: define ESLD
Thank you for this suggestion which will increase clarity for the reader. The piece has been updated throughout using proper terminology.
- Line 147 here is 1,3)-β-d-Glucan used : previously just beta-D-glucan or BDG is used
Thank you for your careful review. We have synchronized terminology throughout the paper.
- line 137-139" Characteristic findings of IPA on chest CT include ground glass opacities, cavities or nodules with or without 138 halo [33]. "Are the CT characteristics different then neutropenic patients?
Thank you for your suggestion which will increase clarity for the reader. This section was rewritten.
- line 105-107 "Despite ongoing efforts toward prevention and improved management of fungal in-105 fection, IC still has significant effects on graft function, morbidity, and mortality post-transplant with reports of 90 day mortality of 26%." In liver transplant recipients or in general?
Thank you for your suggestion which will increase clarity for the reader. This section was rewritten. In specific, we have adjusted to 30% for liver transplant recipients.
- line 146: "(1,3)-β-d-Glucan is less specific in the liver transplant popula-146 tion than other high-risk populations such as hematologic malignancy, allogeneic HSCT 147 and both tests should be interpreted in the complete clinical context [32, 35, 36]. " Why is it less specific
Thank you for your suggestion which will increase clarity for the reader. This section was rewritten.
- line 161 : Here the first time "disseminated" aspergillosis is used.
Thank you for your careful review. This section was rewritten to increase its clarity.
- Line 163. "poor outcome was 2.5 time higher if multiorgan involvement was present."Thus ~85% mortality in multi organ involvement? Please clarify and rephrase
Thank you for your suggestion which will increase clarity for the reader. This section was rewritten.
- line 177 * C. gattii
Thank you for this suggestion. The piece has been updated with the correct nomenclature.
- Paragraph diagnosis, line 209. Questions: cryptococcal antigen on what sample: serum/CSF fluid? Any specific sensitivity/specificity for liver transplant patients. Is the sensitivity of 88-91% for liver transplants only or all SOT? was the lateral flow assay used or the (less sensitive) ELISA?
Thank you for your suggestion which will increase clarity for the reader. This section was rewritten.
- Outcome paragraph cryptococcosis. Nothing is mentioned about outcome. What is known for liver transplant patients.
Thank you for your suggestion which will increase clarity for the reader. This section was rewritten we have included as much specific information to liver transplant recipients as we could to increase clarity.
- zygomycetes should be mucormycetes.
Thank you for this suggestion. The piece has been updated throughout with the correct nomenclature.
- line 247 Should be Scedosporium and Lomentospora based on the reference?
Thank you for your suggestion which will increase clarity for the reader. This sentence was updated per your recommendation.
- Line 250-251 "Fusarium typically manifests as local infection after solid organ transplant with fusariosis being exceedingly rare" Probably more clear to state disseminated fusariosis?
Thank you for your suggestion which will increase clarity for the reader. This sentence was updated per your recommendation.
- Line 257. Scedosporium prolificans (Lomentospora prolificans) should be Lomentospora prolificans (previously Scedosporium prolificans)
Thank you for your careful review. We apologize for this error. It was corrected.
- line 368: liposomal or conventional.
Thank you for your suggestion which will increase clarity for the reader. This sentence was updated per your recommendation.
Reviewer 3 Report
Remarks:
I did not see Authors affiliations.
Not: zygomycetes, but now: mucormycoses.
Last sentence of abstract needs logical correction.
Author Response
Reviewer 3
We sincerely appreciate your careful review. We have addressed your concerns as below.
Remarks:
I did not see Authors affiliations.
Thank you for your suggestion. These have been added.
Not: zygomycetes, but now: mucormycoses.
Thank you for this suggestion. The piece has been updated throughout with the correct nomenclature
Last sentence of abstract needs logical correction.
Thank you for your suggestion which will increase clarity for the reader. This sentence was updated per your recommendation
Round 2
Reviewer 1 Report
The authors appears that they did their best to respond to the comments of the reviewers. Considering the big direction of the article, no further modifications seems to be needed.
Reviewer 2 Report
The manuscript has been adjusted appropriately